# Probing the surface charge of condensates using microelectrophoresis

Merlijn H. I. van Haren ®[1], Brent S. Visser ®[1] & Evan Spruijt ®[1]✉

Biomolecular condensates play an important role in cellular organization. Coacervates are commonly used models that mimic the physicochemical properties of biomolecular condensates. The surface of condensates plays a key role in governing molecular exchange between condensates, accumulation of species at the interface, and the stability of condensates against coalescence. However, most important surface properties, including the surface charge and zeta potential, remain poorly characterized and understood. The zeta potential of coacervates is often measured using laser doppler electrophoresis, which assumes a size-independent electrophoretic mobility. Here, we show that this assumption is incorrect for liquid-like condensates and present an alternative method to study the electrophoretic mobility of coacervates and in vitro condensate models by microelectrophoresis and single-particle tracking. Coacervates have a size-dependent electrophoretic mobility, originating from their fluid nature, from which a well-defined zeta potential is calculated. Interestingly, microelectrophoresis measurements reveal that polylysine chains are enriched at the surface of polylysine/polyaspartic acid complex coacervates, which causes the negatively charged protein α-synuclein to adsorb and accumulate at the interface. Addition of ATP inverts the surface charge, displaces α-synuclein from the surface and may help to suppress its interface-catalyzed aggregation. Together, these findings show how condensate surface charge can be measured and altered, making this microelectrophoresis platform combined with automated single-particle tracking a promising characterization technique for both biomolecular condensates and coacervate protocells.

Condensates formed by liquid-liquid phase separation (LLPS) are important organizers of biomolecular processes in cells and may have played a role as dynamic protocellular compartments during the emergence of cellular life[1–3]. Physicochemical properties and chemical activity of biomolecular condensates (BMCs) and coacervate protocells are well approximated by in vitro systems, such as complex coacervates[4–6]. One property of particular interest in understanding the role of BMCs and coacervate protocells in complex systems is their surface charge. Condensates lack a membrane, but a variety of molecules and particles, including polymers[7], disordered proteins[8–10],

protein clusters[11], liposomes[12] and bacteria[13], have been found to localize to condensate surfaces, possibly mediated by their surface charge. Moreover, the interaction between condensates and lipid membranes, which is partly governed by surface charge, has recently attracted significant attention[14], giving rise to membrane-stabilized coacervate protocells[15], membrane remodeling by condensates[16,17], and coacervate-mediated delivery across membranes via endocytosis or by direct membrane penetration[18,19]. These findings make coacervates potentially interesting drug delivery devices that can transport cargo into cells[20].

[1]Institute for Molecules and Materials, Radboud University, Heyendaalseweg 135, 6523 AJ Nijmegen, The Netherlands. ✉e-mail: e.spruijt@science.ru.nl

All these phenomena depend on the surface charge of condensates. However, the surface charge, or related properties such as the surface potential or zeta potential ($\zeta$-potential), of BMCs is difficult to measure, and reports of coacervate surface charge show a worryingly large variation[7,19,21,22]. The surface charge of colloidal particles is usually determined from their motion in an electric field[23]. Formally, analysis of particle motion in an electric field yields the $\zeta$-potential, which is the electrical potential at the fluid slipping plane near the particle surface. The $\zeta$-potential is slightly lower than the bare surface potential, because of immobile counterions bound to the particle surface in the Stern layer. Nevertheless, the $\zeta$-potential is often considered to be more significant for the stability of colloidal dispersions and droplets. For isolated colloids, the surface potential and $\zeta$-potential can be converted to a surface charge density and an effective surface charge density at the slipping plane, respectively, using Gauss law[24].

Currently, the main technique to determine the $\zeta$-potential of coacervates is laser doppler electrophoresis or electrophoretic light scattering (ELS), in which the frequency shift of a laser beam caused by an oscillating electric field is used to determine the particle mobility, which is usually converted into the $\zeta$-potential or surface charge using the theory of electrophoresis of suspended particles developed by Smoluchowski[25,26].

However, this technique gives fundamentally incorrect results for polydisperse liquid droplets, such as coacervates. The Smoluchowski equation, which relates the electrophoretic mobility of suspended particles, liposomes and polymersomes to their $\zeta$-potential, is not valid for liquid droplets, even if the droplets are very large compared to the characteristic electrostatic screening length, because of their mobile surface charge, the nonrigidity of the interface, and the coupling between hydrodynamic and electrokinetic forces on both sides of the interface[27]. According to the Smoluchowski equation, the electrophoretic mobility of particles is independent of their size when the Debye length is small enough. However, it was recently shown that complex coacervates consisting of polydiallyldimethylammonium chloride (PDDA) and ATP move in electric fields with velocities linearly proportional to their size and the applied electric field strength[28]. Analysis of coacervates by electrophoretic light scattering would thus result in broad $\zeta$-potential distributions that mostly reflect the size distribution of coacervates rather than their true $\zeta$-potential or surface charge.

Additionally, ELS has the disadvantage that the electrophoretic mobility is determined from particle displacements in the vertical direction. Since coacervate droplets and other in vitro models of BMCs are usually in the micrometer size range, they sediment rapidly, giving rise to artifacts in the recorded electrophoretic mobilities[26]. Moreover, coacervates have been shown to exhibit self-organization, internal fusion, vacuolization and other shape instabilities leading to fission at high electric field strengths, which underlines their dynamic nature. Such behavior may also occur in the ELS measurements, further obfuscating the analysis of the coacervate surface charge[29–31].

Here, we report a microelectrophoresis platform to directly quantify the electrophoretic mobility of complex coacervates and model BMCs at a single droplet level to overcome these limitations and determine their $\zeta$-potential. We record the change in position of coacervate droplets in response to an electric field using light microscopy and automated image analysis and calculate the electrophoretic mobility of each individual droplet present in the field of view. The $\zeta$-potential of condensates is calculated using the theory of Ohshima et al. for the electrophoresis of polarizable liquid droplets[32]. Our method can be used in a wide range of conditions and gives direct insights into the effects of environmental conditions, client molecules and biochemical processes on the surface properties of condensates. Microelectrophoresis has previously been used to determine the

electrophoretic mobility of liposomes[33], polymeric nanoparticles[34], silica rods[35] and mammalian cells[36], but until now its application to coacervates was lacking, because of their tendency to coalesce and wet most solid surfaces. Previously, Welsh et al. reported a microfluidic system capable of tracking condensates in an electric field, however their setup did not allow for a correlation between the radius of each single droplet with its electrophoretic mobility[37]. We resolved these limitations by passivating all surfaces with a polyethylene glycol (PEG)-graft-polylysine based polymer that allows for the visualization of all condensates in one plane of focus[38].

By analyzing the mobility of peptide-based complex coacervates of poly-L-lysine (PLL) and poly-L-aspartic acid (PLD), and model BMCs, including nucleophosmin-1 (NPM1)/RNA, our platform revealed several non-trivial features of condensates. All condensates measured have a nonzero surface charge and an electrophoretic mobility that depends linearly on their radius and the inverse Debye length, which has possible implications for their interactions in cells. Disordered proteins that localize to the surface of coacervates, such as α-synuclein, directly alter the surface charge, providing mechanisms underlying interfacial localization. Interestingly, α-synuclein can be completely displaced from the coacervate surface by addition of ATP, which may help to suppress surface-mediated aggregation. ATP can invert the surface charge of condensates, suggesting it can be enriched at the coacervate surface. These findings shed light on condensate features and the role of intracellular metabolites, such as ATP, in controlling condensate function. In short, microelectrophoresis of coacervates at the single droplet level holds great promise in determining fundamental physicochemical properties of condensates and predicting their interactions with other solutes.

## Results and discussion
### Electrophoretic mobility of complex coacervates
To investigate the effect of an electric field on coacervate droplets, we connected a direct current power source (6 V) to a microchannel containing PLL/PLD ($K_{10}/D_{10}$) coacervate droplets. The coverslip was passivated to prevent coacervate droplets from wetting the glass, which allows the droplets to retain their spherical shape (Fig. 1a)[38]. In the absence of an electric field, droplets slowly coalesced due to gravitational settling and Brownian motion. When the electric field was applied, droplets immediately started to migrate in a straight trajectory in the direction of the cathode (Fig. 1b, Supplementary Movie 1). During their movement, a fraction of coacervates coalesced to form larger droplets, that continued in the same direction as their ancestral droplets, but with a higher velocity.

When analyzing the droplet traces in more detail, we noticed that large droplets have a higher velocity than small droplets (Fig. 1c). A plot of the coacervate velocity as a function of their radius revealed a linear relationship (Fig. 1d), similar to what has been shown before for polyelectrolyte coacervate droplets that were stabilized by quenching them in deionized water[28]. We note that such a size-dependent mobility is not predicted by Smoluchowski's classical theory of electrophoresis of suspended particles (Supplementary Information), which is commonly used to analyze electrophoretic light scattering (ELS) measurements (performed on a zetasizer, for instance), including measurements of coacervate droplets[7,19,21,22]. This theory is not valid for liquid droplets and its application to ELS gives fundamentally incorrect results for the zeta potential ($\zeta$-potential)[27].

To describe the size-dependent electrophoretic mobility of coacervate droplets, we used the theoretical framework derived by Ohshima et al. for the motion of polarizable liquid droplets in an electrolyte solution[32]. For large values of droplet radius ($R$) compared to the Debye screening length, $\kappa^{-1}$ ($\kappa R \gg 1$), and a droplet viscosity ($\eta_d$) that is significantly larger than the viscosity of the surrounding

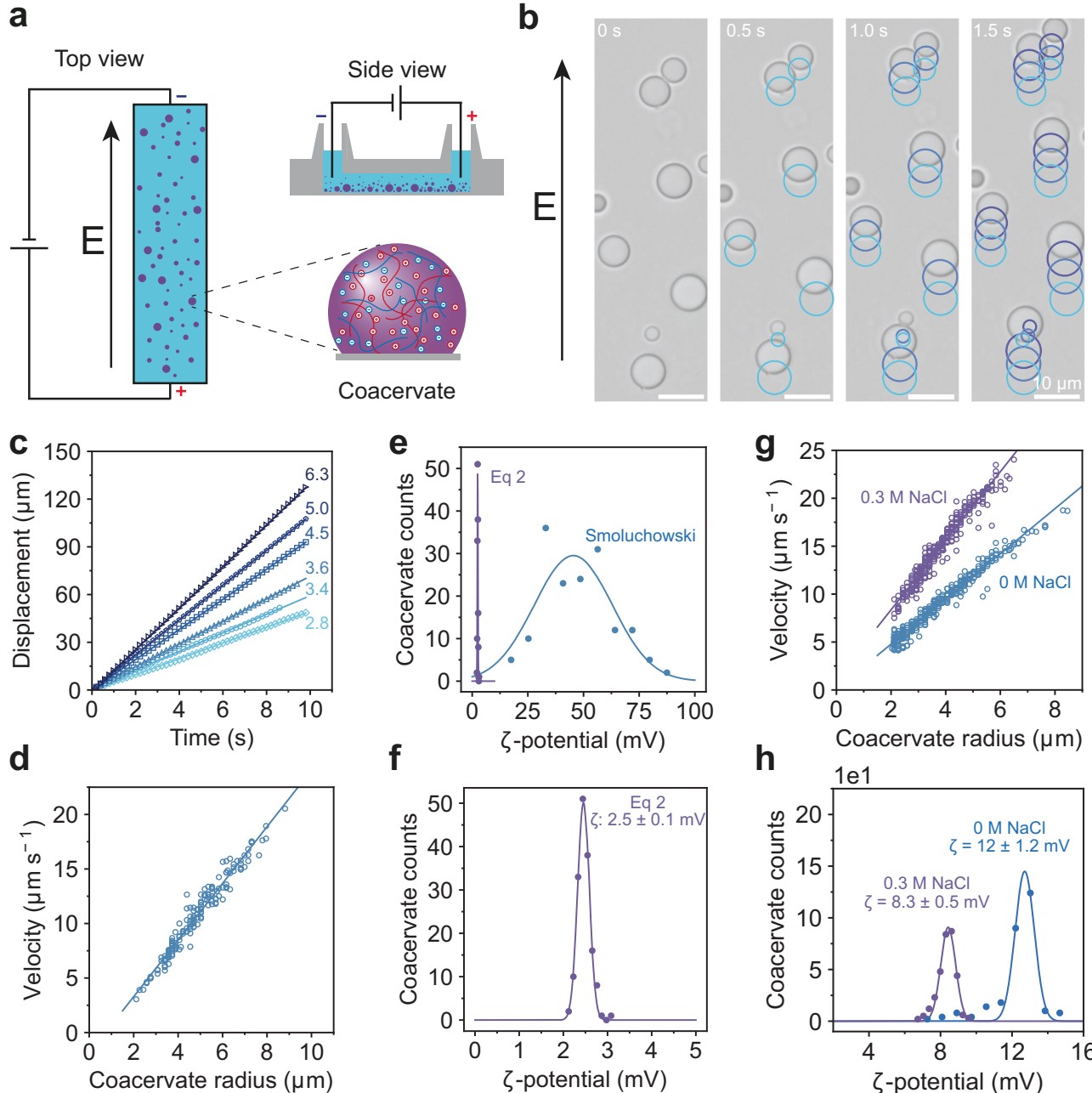

**Fig. 1 | Microelectrophoresis of coacervates shows a size-dependent velocity.**
**a** Experimental design of the microelectrophoresis setup for coacervates, consisting of a μ-channel slide containing electrodes connected to a direct current power source. **b** Brightfield microscopy images of $K_{10}/D_{10}$ coacervates in an electric field of 3.84 V cm$^{-1}$, the droplets move towards the cathode. **c** Displacement of $K_{10}/D_{10}$ coacervates over time, number at droplet trace indicates its radius in μm. **d** Average velocity of $K_{10}/D_{10}$ coacervates ($n = 160$ individual droplets) in an electric field of 3.84 V cm$^{-1}$ plotted against their radius. The measured velocities were fitted to a linear equation. **e** ζ-potential distributions of $K_{10}/D_{10}$ coacervates determined by the Smoluchowski equation and Eq. 2. **f** A closer look at the ζ-potential distribution obtained from Eq. 2. **g** Electrophoretic mobility of $K_{30}/D_{30}$ coacervates at 0 M ($n = 369$ individual droplets) and 0.30 M ($n = 314$ individual droplets) sodium chloride. **h** ζ-potential distributions of $K_{30}/D_{30}$ droplets determined by Eq. 2. The following parameters were used for calculations: $\eta_s = 0.89$ mPa·s, $\eta_d = 37$ mPa·s and $\kappa^{-1} = 1.4$ nm for $K_{10}/D_{10}$, $\eta_d = 130$ mPa·s and $\kappa^{-1} = 1.4$ nm for $K_{30}/D_{30}$ coacervates, and $\eta_d = 110$ mPa·s and $\kappa^{-1} = 0.5$ nm for 0.30 M added NaCl. Coacervates consist of 5 mM $K_n$ and $D_n$ (monomer concentrations) in 50 mM Tris pH 7.4 buffer. Velocities (**d**, **g**) were fitted with a linear equation, and the slope was used to determine the ζ-potential, as described in the Supporting information. Distributions in (**e**, **f**, **h**) are fitted with a Gaussian probability density function, and the mean μ and standard deviation σ are shown in the plots. An overview of the fits, sample sizes and statistics can be found in Supplementary Table 3. Source data are provided as a Source Data file.

solution ($\eta_d/\eta_s = \eta_r \gg 1$), we obtained the following direct relation (Supplementary Information and Supplementary Figs. 1 to 5):

$$v = \frac{\varepsilon_r \varepsilon_0 \zeta E}{\eta_s}\left(1 + \frac{\kappa R}{3\eta_r}\right) \qquad (1)$$

where $v$ is the droplet velocity, $\varepsilon_r$ the dielectric constant, $\varepsilon_0$ the vacuum permittivity, $\zeta$ the zeta-potential, $E$ the electric field strength, $\eta_s$ the viscosity of the medium, $\kappa R$ the coacervate radius divided by the Debye length and $\eta_r$ the viscosity ratio of the droplet to the surrounding solution phase. Equation 1 resembles the Smoluchowski equation but contains an extra term that contains the droplet size and the viscosity

ratio $\eta_r$. For solid particles, the viscosity ratio diverges, and in this limit, the Smoluchowski equation is retrieved. However, condensates and coacervates have a finite viscosity, which results in a size-dependent velocity, as observed for $K_{10}/D_{10}$ coacervates (Fig. 1d). Equation 1 can be rewritten to calculate the ζ-potential from the velocity of moving droplets.

$$\zeta = \frac{3\eta_d v}{\varepsilon_r \varepsilon_0 E} \left( \frac{1}{3\eta_r + \kappa R} \right) \qquad (2)$$

When calculating the ζ-potential of coacervates, their velocity, radius and viscosity should be accurately determined. With our microelectrophoresis setup (Supplementary Fig. 6), the coacervate velocity and radius are recorded for each individual droplet by the particle tracking algorithm, whereas the electric field strength and Debye length are known experimental parameters (Supplementary Fig. 7). Determining the viscosity of coacervates and condensates can be challenging, as they exhibit an elastic response to rapid deformations[39–41], and classical rheometry of a bulk coacervate phase requires a substantial amount of material. Here, we used raster image correlation spectroscopy (RICS) to measure the diffusion coefficient of rhodamine labeled dextran and calculate the condensate viscosity using the Stokes-Einstein relation, and found a viscosity of 37 mPa·s for $K_{10}/D_{10}$ coacervates (Supplementary Fig. 9)[42]. The ζ-potential of $K_{10}/D_{10}$ droplets was thus calculated to be +2.5 ± 0.1 mV, and is narrowly distributed (Fig. 1e, f). It is striking to compare this distribution with the ζ-potential distribution obtained when applying the classical Smoluchowski equation for electrophoretic mobility of solid particles. The latter results in a very wide distribution of ζ-potentials, because the mobility is assumed to be independent of droplet size (Fig. 1e).

According to the theory of Ohshima et al., the electrophoretic mobility is not only dependent on the particle size, but also on the effect of screening through the electrical double layer[32]. The influence of $\kappa^{-1}$ was studied by measuring the velocities of $K_{30}/D_{30}$ coacervates with two different concentrations of sodium chloride added. Coacervates with 0 and 0.30 M sodium chloride added, have a $\kappa^{-1}$ of 1.4 and 0.5 nm, respectively. In both samples, the coacervates are significantly larger than the screening length ($\kappa R \gg 1$), but should still have different mobilities according to Eq. 1. Indeed, $K_{30}/D_{30}$ coacervates containing 0.30 M sodium chloride moved significantly faster than coacervates without additional salt (Fig. 1g). Additionally, the viscosity of $K_{30}/D_{30}$ coacervates decreased from 130 to 110 mPa·s[43]. Nevertheless, the ζ-potential of $K_{30}/D_{30}$ coacervates, which takes the change in $\kappa^{-1}$ and $\eta_d$ into account, is +12 mV for no salt and +8.3 mV for 0.30 M salt (Fig. 1h). The decrease in surface potential is expected for droplets with a constant surface charge density when the screening length is decreased at higher salt concentration[44]. When the ζ-potential is measured with laser doppler electrophoresis, which uses the Smoluchowski equation to convert mobility to ζ-potential, we find ζ-potentials of 49 and 85 mV. It is unlikely that a higher salt concentration would significantly increase the coacervate ζ-potential, as the increased ionic strength would lead to charge screening and would ultimately lower the surface potential[44]. These observations emphasize the importance of using microelectrophoresis to determine meaningful ζ-potentials of coacervate droplets.

To exclude the possibility that the observed motion is significantly influenced by the electroosmotic flow in the µ-channel, we determined the electrophoretic mobility of $K_{10}/D_{10}$ coacervates containing 10 µM Alexa488 at a z-position of 100 µm inside the capillary above the glass surface. Velocities of coacervates moving through the middle of the channel did not differ significantly from coacervates moving directly over the surface of the glass slide, indicating that electroosmotic flow is negligible for this microelectrophoresis platform (Supplementary Fig. 10). In addition, the electrophoretic mobility of negatively charged polystyrene beads with a known ζ-potential of

−6.7 ± 0.6 mV (determined by ELS) was measured in the microelectrophoresis setup. These beads moved with a constant velocity towards the anode when an electric field was applied and a ζ-potential of −6.0 ± 0.1 mV was calculated, underscoring the validity of our approach (Supplementary Fig. 11).

## Complex coacervate surface potential of peptide coacervates depends on composition

After having established a reliable method to determine the ζ-potential of coacervates, we decided to investigate the effect of coacervate composition on their ζ-potentials. We prepared coacervates of poly-L-lysine(PLL)/poly-L-aspartic acid (PLD) with three different lengths ($K_{10}$, $K_{30}$, $K_{100}$ and $D_{10}$, $D_{30}$, $D_{100}$) at equivalent monomer concentrations and measured their mobility at different field strengths. All coacervates showed a radius-dependent electrophoretic mobility (Fig. 2a). The ζ-potential of the $K_n/D_n$ coacervates was determined by fitting the electrophoretic mobility to the first derivative of Eq. 2 (Fig. 2b, c), with coacervate viscosities obtained by RICS (Supplementary Table 1). $K_{30}/D_{10}$ coacervates have the highest ζ-potential of the measured compositions. It is often assumed that a relatively large length difference between coacervate components would shift the coacervate ζ-potential to the value of the longest species, as shorter chains have a larger entropic gain per residue when they dissolve from the coacervate into the surrounding solution[45]. However, we find that the chemical characteristics of the coacervate components play a more important role in determining the surface properties. In the case of $K_n/D_n$ coacervates, the lysines tend to remain incorporated more than aspartic acids, thus creating a net positive surface charge at equal length: $K_{10}/D_{10}$ and $K_{30}/D_{30}$ coacervates are both positively charged, while the PLD is of the same length. Compared to other studies, we do not find ζ-potentials exceeding 20 mV[37,46,47]. The reason for these lower ζ-potentials is that particles with a finite viscosity have a significantly higher electrophoretic mobility than solid particles[32]. This, together with coacervates' tendency to rapidly coalesce, which would be somewhat hindered by a high surface charge, leads us to hypothesize that many coacervate ζ-potentials are significantly lower than previously reported[37].

$K_{10}/D_{30}$, $K_{30}/D_{100}$ and $K_{100}/D_{100}$ coacervates all showed a very low mobility, suggesting that their surface is near neutral. To confirm that the lack of mobility was the result of a low ζ-potential, and not from droplets adhering to the µ-channel surface, the electrophoretic mobility of $K_{10}/D_{30}$ coacervates was measured at varying monomer compositions (ratios of 2:1, 1:1 and 1:2). At concentration ratios of 2:1 and 1:2 $K_{10}$ to $D_{30}$, the coacervates showed a radius-dependent electrophoretic mobility and were charged, with a ζ-potential of +8.6 ± 1.3 mV and −4.4 ± 0.2 mV, respectively (Fig. 2d). It is interesting to see how simply changing the relative concentrations of components in a coacervate can reverse their surface charge, and thereby affect their affinity for membranes and certain surface-localizing proteins. These findings could have implications for the functioning of BMCs in cells[48].

## A variety of condensates have nonzero surface charges

To investigate whether the size-dependent electrophoretic mobility is a general property of liquid condensates, we studied the behavior of four condensate systems often used for in vitro BMC studies with our microelectrophoresis setup. The surface charge of condensates has not been studied in detail so far. Nonetheless, the sign and magnitude of a biomolecular condensate's surface charge could have important implications in cells, such as regulating the affinity to negatively charged membranes or its interaction with client molecules and protein clusters[11,19,49].

We prepared condensates consisting of a disordered elastin-like peptide (ELP), $K_{72}$, N-terminally labeled with GFP, and ATP[38], a ribonucleoprotein (RNP) condensate model system consisting of an

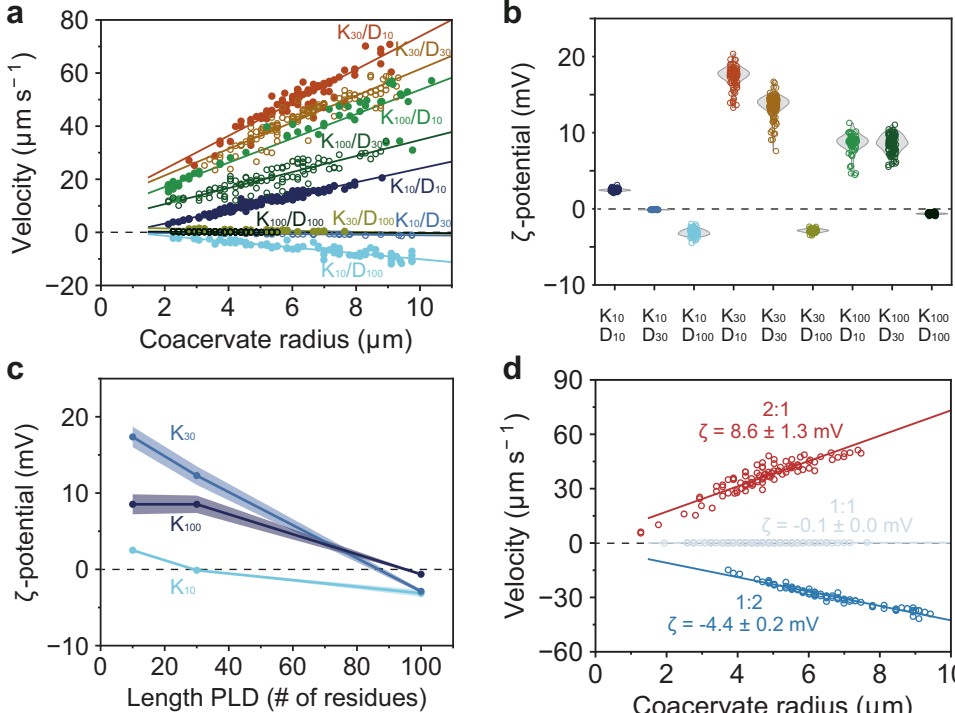

**Fig. 2 | ζ-potentials of lysine/aspartic acid coacervates determined by micro-electrophoresis. a** size-dependent electrophoretic mobility of $K_n/D_n$ coacervates in an electric field of 3.84 V cm$^{-1}$. **b, c** ζ-potentials of equimolar $K_n/D_n$ coacervates calculated by fitting the first derivative of Eq. 2 to coacervate velocities. Line represents the mean ζ-potential and shaded regions indicate the standard deviation shown in (**b**) ($n > 30$ individual droplets). **d** Changing mixing ratio of $K_{10}/D_{30}$ droplets recovers electrophoretic mobility and changes the ζ-potential. Coacervates consist of 5 mM $K_n$ and $D_n$ (monomer concentrations) in 50 mM Tris pH 7.4 buffer. Parameters used to calculate ζ-potentials can be found in Supplementary Table 2. Velocities (**a, d**) were fitted with a linear equation, and the slope was used to determine the ζ-potential, as described in the Supporting information. An overview of the fits, sample sizes and statistics can be found in Supplementary Table 3. Source data are provided as a Source Data file.

arginine-rich peptide with sequence [RGRGG]₅ (RRP) and RNA (polyU), a condensate model of the nucleolus granular component consisting of nucleophosmin-1 (NPM1) and RNA, and condensates consisting of a highly charged protamine and NADH. Condensates formed by the ELP and ATP were positively charged, as the cationic ELP is the largest component by orders of magnitude (0.5 kDa for ATP and 64 kDa for $K_{72}$, Fig. 3a). RNP-like condensates formed by an arginine-rich peptide and RNA were negatively charged (Fig. 3b). Interestingly, these coacervates did not show size-dependent electrophoretic mobility in our usual buffer concentration, but they did when sodium chloride was added, suggesting that these condensates behave like elastic gel particles at low ionic strength, in agreement with previous reports[39,50]. The addition of sodium chloride induces liquefaction and reduces the viscosity of RRP/polyU condensates, which results in an emphasis on the size-dependent term of Eq. 2. Nucleolus-like condensates formed by NPM1 and RNA were negatively charged, as both NPM1 and polyU are negatively charged (Fig. 3c). These condensates also exhibited a size-dependent mobility. Finally, highly charged protamine-based condensates were positively charged and also moved with size-dependent velocity (Fig. 3d). A well-defined ζ-potential was calculated for each condensate system, ranging from −11 mV to +1.9 mV. Taken together, these examples show that a wide range of liquid condensates exhibit charged interfaces, which could have implications for the interactions with cellular content, such as proteins, organelles and other condensates present. Condensates with a higher ζ-potential could be more stable against fusion, as predicted by classical emulsion theory[37,51]. This is reflected by the relatively larger size of the ELP/ATP condensates, as their radius is approximately twice as large after 1 h of incubation compared to the other condensates, while bearing the lowest net surface charge.

## Client molecules can alter coacervate surface charge and limit protein adsorption

An advantage of our microelectrophoresis setup is that it allows direct evaluation of the effect of specific proteins, peptides and other solutes on the surface properties of condensates. When studying the partitioning or adsorption of guest molecules to condensates, fluorescently labeled molecules are often required, which is not always feasible and may influence the molecular properties. To illustrate the potential of the microelectrophoresis platform described here, we set out to investigate what the effect common metabolites, such as ATP, have on the coacervate surface charge. Firstly, ATP was added in a stepwise manner to positively charged $K_{10}/D_{10}$ droplets. Interestingly, we found that addition of 1 mM ATP resulted in a complete suppression of the electrophoretic mobility of the coacervates, suggesting that their surface charge is nearly zero due to the adsorption of ATP to the interface. We note that ATP likely also mixes with the coacervate interior, displacing some of the $D_{10}$, thus changing its composition. However, to suppress the surface charge, a surface excess of ATP is required. Indeed, when more ATP was added to $K_{10}/D_{10}$ coacervates, the electrophoretic mobility was recovered, but the sign of the coacervate surface charge was inverted (Fig. 4a, b). The fact that addition of a small common metabolite such as ATP can have a strong effect on coacervate surface properties may be relevant in the context of condensates, as ATP is actively regulated in cells, and ATP levels may vary under stress conditions. In turn, this could affect the partitioning of proteins and other biomolecular guests[52].

We wondered if the surface modulation by ATP could also modulate the adsorption of other species to condensate interfaces. Besides providing energy for biochemical reactions in cells, it was found that ATP can act as a hydrotrope that can help solubilize proteins and

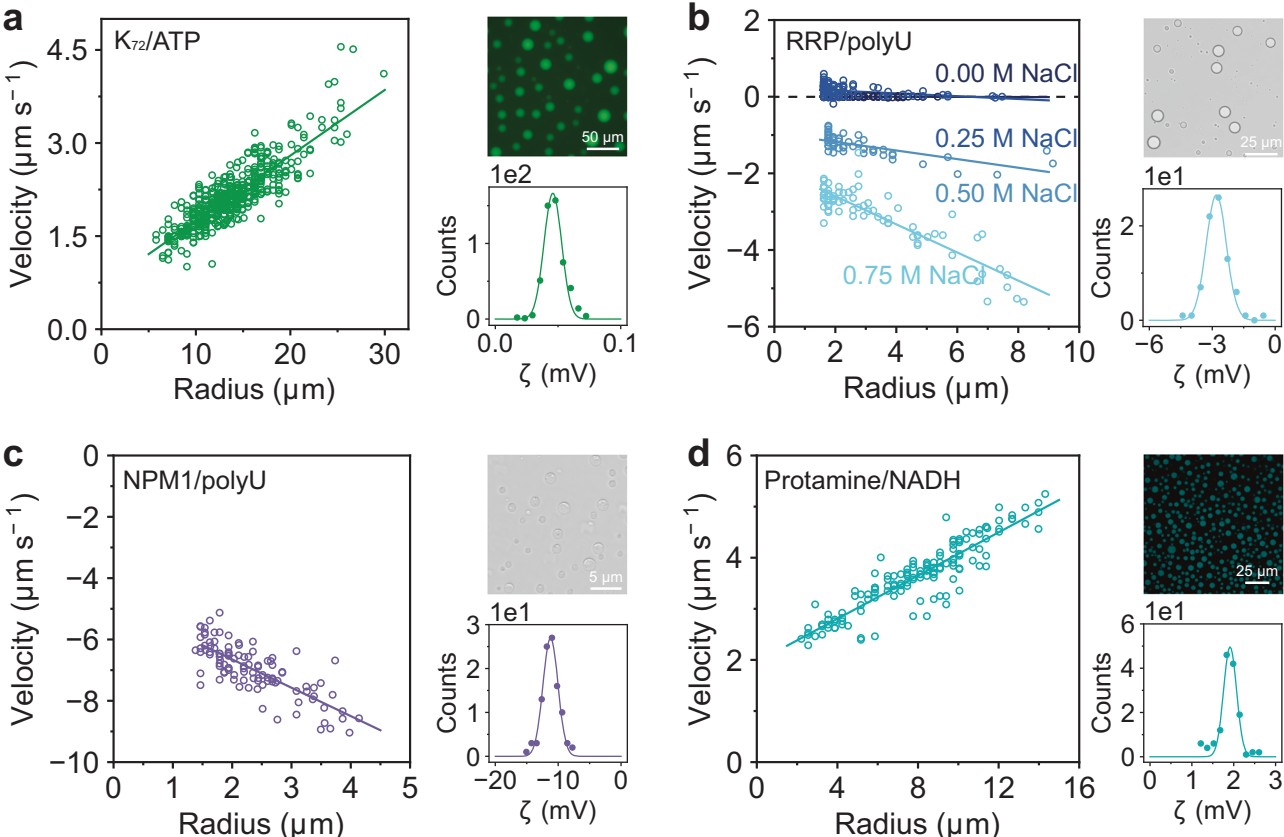

**Fig. 3 | Various in vitro condensates show size-dependent electrophoretic mobility. a** Electrophoretic mobility and fluorescence microscopy of GFP labeled $K_{72}$/ATP coacervates. **b** RRP/polyU are too viscous to display electrophoretic mobility at 3.75 V cm$^{-1}$, but do when their viscosity is reduced by addition of sodium chloride. **c** Electrophoretic mobility of NPM1/polyU condensates, these condensates are negatively charged. **d** Electrophoretic mobility and fluorescence microscopy of protamine/NADH coacervates. Parameters used to calculate ζ-potentials can be found in Supplementary Table 2. Velocities were fitted with a linear equation, and the slope was used to determine the ζ-potential, as described in the Supporting Information. The ζ-potential distributions were fitted with a Gaussian probability density function, and the mean μ and standard deviation σ are reported in Supplementary Table 2. An overview of the fits, sample sizes and statistics can be found in Supplementary Table 3. Source data are provided as a Source Data file.

influence phase separation of biomolecular condensates[53,54]. Recently, it was shown that α-synuclein adsorbs to the interface of poly-DL-lysine (pLys)/poly-DL-glutamic acid (pGlu) droplets, causing a substantial enhancement of its rate of aggregation[9]. To determine the effect of α-synuclein on coacervate surface properties, we measured the electrophoretic mobility of pLys/pGlu coacervates in the absence and presence of α-synuclein. In the absence of α-synuclein present, pLys/pGlu coacervates displayed size-dependent electrophoretic mobility and had a positive surface potential (Fig. 4c). After addition of 1 μM α-synuclein, droplets no longer responded to the electric field, indicating that their positive surface charge was neutralized by the negatively charged α-synuclein that is adsorbed on the droplet interface (Fig. 4c). Further addition of α-synuclein did not invert the surface charge of pLys/pGlu coacervates, suggesting that the surface is saturated with α-synuclein.

When 3 mM ATP was added to α-synuclein containing pLys/pGlu coacervates, the coacervates started to move again, albeit in the direction of the anode, indicating that their surface acquired a negative charge (Fig. 4c). At the same time, the fluorescently labeled α-synuclein was completely displaced from the droplet interface and released into solution (Fig. 4d–f). We note that the trend of the electrophoretic mobility of the coacervates as a function of their size after displacement of α-synuclein by ATP is reversed compared to other experiments, which precludes direct determination of their ζ-potential by Eq. 2. We suspect that the reason for this is the presence of small amounts of α-synuclein clusters or oligomers that were not displaced

by ATP on the surface of pLys/pGlu droplets, as can be seen upon close inspection in Fig. 4e. Nevertheless, these findings of α-synuclein displacement are relevant in the context of protein aggregation, as α-synuclein and other proteins have previously been found to exhibit enhanced aggregation at the surface of condensates[9,10,55]. Seeing how both proteins and metabolites have an effect on the coacervate surface charge raises questions about the state of the BMC surfaces in cells, as the cellular cytoplasm is filled with metabolites and proteins that could partition or adsorb onto condensate interfaces[11]. The microelectrophoresis platform presented here enables a quantitative and systematic investigation of the effect of different proteins and metabolites on condensate surface properties, thus shedding light on their localization and possible role in condensate homeostasis.

In conclusion, coacervate droplets are interesting and versatile in vitro model systems for protocells and biomolecular condensates. Here, we show a microelectrophoresis platform to determine the electrophoretic mobility of coacervates and other biomolecular condensates by tracking hundreds of individual droplets in an electric field of 0 to 15 V cm$^{-1}$. Coacervate droplets showed a size-dependent velocity, with large droplets moving faster than small droplets in the direction of the electric field. These observations ask for a reconsideration of the use of the Smoluchowski equation, which is often used to calculate the zeta potential of coacervates. We derive a simplified equation that describes the size-dependent mobility of coacervates, based on the theory of Ohshima et al. The electrophoretic mobility of coacervates is not only dependent on their zeta potential, as is

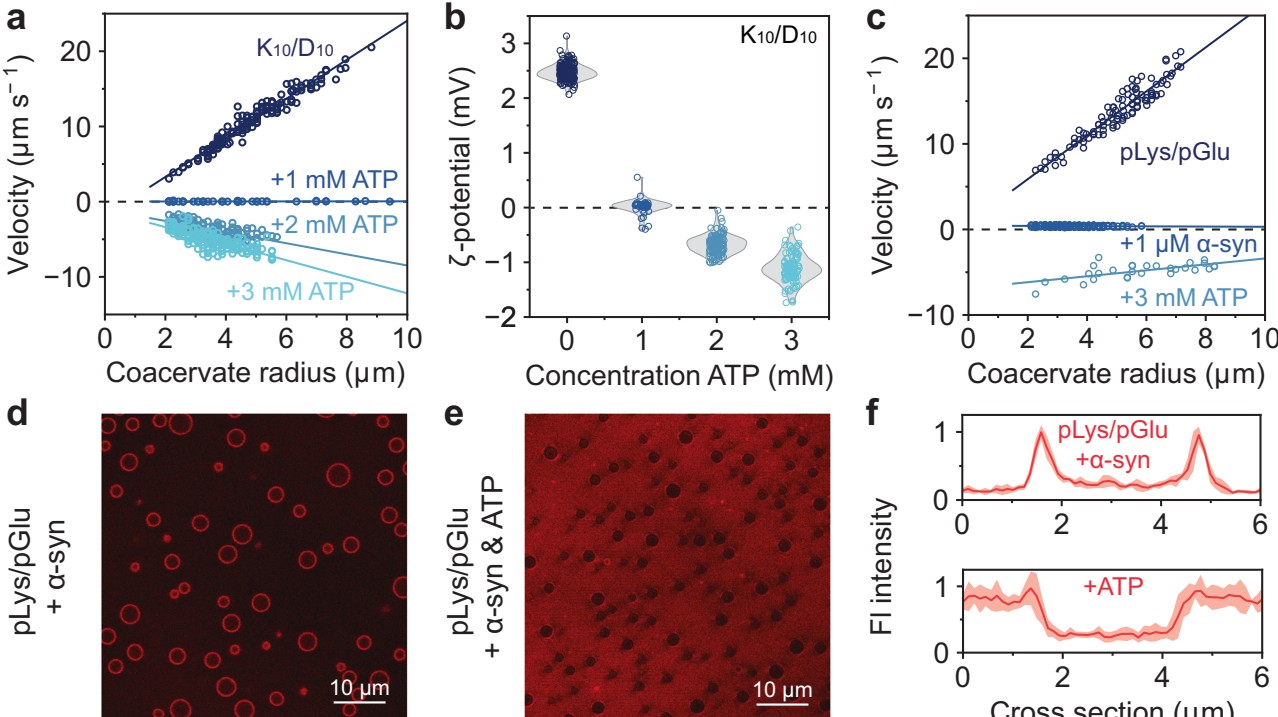

**Fig. 4 | Addition of ATP and α-synuclein to peptide coacervates alters their surface potential. a** The positive surface charge of $K_{10}/D_{10}$ droplets is inverted when adding negatively charged ATP. **b** ζ-potentials of $K_{10}/D_{10}$ droplets with increasing amounts of ATP added. **c** The electrophoretic mobility of pLys/pGlu coacervates disappears when α-synuclein adsorbs to their surface, masking the positive surface charge. The surface charge is recovered, but inverted when ATP is added. **d** pLys/pGlu coacervates with α-synuclein (Alexa 647 labeled) adsorbed on the interface. **e** Addition of ATP removes α-synuclein from the coacervate interface. **f** Mean fluorescence intensity (red line) of α-synuclein on pLys/pGlu droplets before (top) and after (bottom) addition of ATP, shaded region represents the standard deviation ($n = 5$ individual droplets). Coacervates consist of 2.4 mM pLys and pGlu (monomer concentrations) in 50 mM Tris pH 7.4 buffer. Parameters used to calculate ζ-potentials can be found in Supplementary Table 2. Velocities (**a, c**) were fitted with a linear curve equation. An overview of the fits, sample sizes and statistics can be found in Supplementary Table 3. Source data are provided as a Source Data file.

assumed in the Smoluchowski equation, but also on their radius, Debye length and internal viscosity. Using this platform, we show that coacervate droplets have a narrowly distributed zeta potential, in sharp contrast to results from electrophoretic light scattering that make use of the Smoluchowski equation.

With these findings, we derived the ζ-potential of different peptide-based coacervates and a variety of model biomolecular condensates and looked at the influence on partitioning of guest molecules on this potential. ATP could reverse the ζ-potential of positively charged coacervates composed of much larger species, presumably by adsorbing to the interface. Interestingly, ATP could also displace α-synuclein from the surface of coacervates, and possibly reduce the interface-catalyzed aggregation that has been previously observed. With the current microelectrophoresis setup, zeta potentials of a wide range of condensates can be determined more accurately than with previously used techniques. This paves the way for a better understanding of condensate properties and the interactions of condensates with other cellular components.

## Methods
### Materials
All chemicals and reagents were used as received from commercial suppliers unless stated otherwise. Milli-Q water (18.2 MΩ cm) from Millipore Corporation was used. The following chemicals were purchased from Sigma Aldrich: sodium chloride, 1.0 M hydrochloric acid, 1.0 M sodium hydroxide, Tris base, adenosine triphosphate disodium salt, polyuridylic acid potassium salt, protamine sulfate (grade X from salmon), L-glutamic acid potassium salt monohydrate and tetramethylrhodamine isothiocyanate–Dextran (MW = 4400 Da). HEPES

free acid was purchased from FluoroChem. Nicotinamide adenine dinucleotide disodium salt trihydrate (reduced, 98%) was purchased from Fisher Scientific. PLL-*g*[3.5]-PEG was purchased from SuSoS. The following oligopeptides were purchased from Alamanda Polymers: poly-L-lysine hydrobromide (MW = 2100 Da, 10-mer), poly-L-lysine hydrobromide (MW = 6300 Da, 30-mer), poly-L-lysine hydrobromide (MW = 21,000 Da, 100-mer) poly-L-aspartic acid sodium salt (MW = 1400 Da, 10-mer), poly-L-aspartic acid sodium salt (MW = 4100 Da, 30-mer), poly-L-aspartic acid sodium salt (MW = 14,000 Da, 100-mer), poly-L-arginine hydrochloride (MW = 1900 Da, 10-mer), poly-D,L-lysine hydrobromide (MW = 21,000 Da, 100-mer), poly-D,L-glutamic acid sodium salt (MW = 15,000 Da, 100-mer), the polydispersity index (PDI) of all peptides was 1.0–1.2. Arginine-rich polypeptide RRP (full sequence [RGRGG]₅) was purchased from Genscript Biotech (The Netherlands, The Hague). GFP-labeled $K_{72}$ and nucleophosmin-1 were expressed and purified as previously described[38,56]. Full-length α-synuclein was expressed, purified and labeled as previously described[9].

### Coacervate formation
All experiments with polylysine (PLL)/polyaspartic acid (PLD) coacervates were performed using Tris buffer (final concentration 50 mM, pH 7.4), by adding PLL to the buffer, followed by PLD, both with a final concentration of 5 mM monomer units. For RRP/polyU coacervates, droplets were made by adding polyU (final concentration 0.38 mg mL⁻¹) and RRP (final concentration 0.5 mM) to the standard base buffer. Sodium chloride was added to reach final NaCl concentrations of 0.25, 0.50 and 0.75 M. $K_{72}$/ATP coacervates were prepared in HEPES buffer (final concentration 50 mM, pH 7.4), by adding

GFP-labeled $K_{72}$ (final concentration 20 μM) to HEPES buffer followed by ATP (final concentration 3 mM). For protamine/NADH coacervates, droplets were made by adding protamine (final concentration 0.4 mM) to Tris buffer, followed by NADH (final concentration 10 mM). NPM1/polyU condensates were prepared by adding NPM1 (final concentration 30 μM) to Tris buffer (20 mM, pH 7.2) containing 0.25 M potassium glutamate, followed by polyU (final concentration 0.1 mg mL$^{-1}$). The PLys/pGlu coacervates were made by adding poly-D,L-lysine (final concentration 2.4 mM) to Tris buffer, followed by poly-D,L-glutamic acid (final concentration 2.4 mM). α-synuclein was always added last to a final concentration of 1–10 μM. All samples for microelectrophoresis measurements were prepared to a total volume of 110 μL in microcentrifuge tubes. Mixing was done by vortexing for 10 s at 2800 rpm (lab dancer, VWR).

### Microscopy
All samples were imaged on μ-Slides VI 0.4 (polymer coverslip, Ibidi) that were cleaned with a plasma cleaner, incubated for 24 h with 0.1 mg mL$^{-1}$ PLL-g[3.5]-PEG (SuSoS, Dübendorf, Switzerland) dissolved in 10 mM HEPES pH 8.0, and washed and dried with Milli-Q water and pressurized air, respectively. Before image acquisition, 100 μL coacervate suspension was transferred to the channel and incubated for 1 h to allow droplets to coalesce and settle on the glass surface. Electrodes (2 mm, silver) connected with copper wires to a BT-305A PSU direct current power source (Basetech) were lowered into the opposing ends of the microchannel and an electric field of 1.2 to 12 V cm$^{-1}$ was applied, with the cathode at the top of the field of view (Supplementary Fig. 6). Moving coacervates were imaged in the middle of the channel of the microslide. Samples were imaged on an Olympus IX83 inverted fluorescence microscope equipped with a motorized stage (TANGO, Märzhäuser) and LED light source (pE-4000 CoolLED). Images were recorded with a 40× universal plan fluorite objective (WD 0.51 mm, NA 0.75, Olympus) with a temperature-controlled CMOS camera (Hamamatsu Orca-Flash 4.0). α-synuclein localization was studied with a Leica SP8x confocal inverted microscope equipped with a DMi8 CS motorized stage, a pulsed white light laser, two HyD SP GaAsP and two PMT detectors. Images were recorded using a 100x HC PL APO oil immersion objective, the laser was set to 633 nm and emission was collected between 650 nm and 750 nm using the HyD detector.

### Laser doppler zeta potential measurements
Zeta potentials of coacervates were measured on a DLS-Zetasizer (Malvern). The same peptide and buffer concentrations were used for surface charge measurements as for microscopy experiments. After coacervates were formed, 1 mL sample was injected into a disposable folder capillary cell (DTS1070) and measured at 25 °C. Samples were diluted when they appeared too turbid. Three measurements per sample were taken, each consisting of 100 scans (Supplementary Fig. 12).

### Condensate viscosity by Raster image correlation spectroscopy (RICS)
The viscosity of condensates was determined using Raster Image Correlation Spectroscopy (RICS) on a Leica SP8 confocal microscope equipped with a single-photon detector. Calibration of the focal volume waist $\omega_0$ was performed using the known diffusion coefficient of Alexa 488 of 435 μm$^2$ s$^{-1}$ ($T = 22.5 \pm 0.5$ °C) in water, and $\omega_z$ was set to 3 times the value of $\omega_0$[57]. All measurements were captured at a resolution of 256 × 256 pixels with a 20 nm pixel size using a 63x objective. Coacervate samples were measured at 10 Hz line speed with 15 frames acquired per data point. Analysis of autocorrelation curves was done using PAM[58]. Measured diffusion coefficients for the labels used for each condensate system and their respective viscosities can be found in Supplementary Table 1.

### Droplet tracking
Raw microscopy videos were processed and analyzed with MATLAB 2021 Image Processing Toolbox. Coacervates were detected and labeled with an algorithm that finds dark and bright contrasted circles, and objects at the edges that were not entirely inside the field of view were removed[59]. The area, perimeter, and centroid of each droplet were extracted and droplets overlapping each other or in close proximity to each other were removed by calculating the circularity (4πA/perimeter$^2$) of every object and removing objects below a threshold circularity value of 0.9. Subsequently, the positions of all remaining droplets were compiled in a list and droplet motion was tracked using a custom tracking script.

The mean velocity of the droplets was calculated from the droplet traces by taking the difference of the y coordinate from the centroid of the droplet between the first and last recorded position and dividing it by the elapsed time. Acceleration and deceleration of the droplets were calculated by taking the difference between velocity of the droplet at every recorded time point. In the analysis, droplets that did not move ($v < 0.5$ μm s$^{-1}$), or suddenly slowed down or sped up (caused by imperfections in the surface modification) were discarded. The fraction of moving droplets (when ζ-potential is nonzero) was always higher than 80%, and is reported for each experiment (Supplementary Table 2).

### Reporting summary
Further information on research design is available in the Nature Portfolio Reporting Summary linked to this article.

## Data availability
The source data generated in this study have been deposited in the Radboud Repository (https://data.ru.nl) under accession code https://doi.org/10.34973/yrs1-et97. The data are available under CC-BY-4.0 license. Source data are provided with this paper.

## Code availability
The code generated and used in this study have been deposited in the Radboud Repository (https://data.ru.nl) under accession code https://doi.org/10.34973/yrs1-et97, and are available under CC-BY-4.0 license.

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

## Acknowledgements

This project received funding from the European Research Council (ERC) under the European Union's Horizon 2020 research and innovation program under grant agreement number 851963, and a Vidi grant from the Netherlands Organization for Scientific Research (NWO). The authors want to thank Remco Fokkink and Wageningen Research University for their assistance and hospitality when measuring zeta potentials using their Malvern Zetasizer. In addition, we would like to thank Alain A. M. André for expression and purification of nucleophosmin-1, and Wojciech P. Lipiński for help with α-synuclein measurements and fruitful discussions.

## Author contributions

E.S. designed and supervised the project. M.H.I.v.H. designed, performed, and analyzed coacervate microelectrophoresis experiments. B.S.V. performed and analyzed viscosity measurements with RICS. M.H.I.v.H. and E.S. wrote the manuscript, with input and revisions from all authors.

## Competing interests

The authors declare no competing interests.
