## [Peer Review File · Nature Communications]

Reviewers' Comments:

Reviewer #1:

Remarks to the Author:

The manuscript by van Haren and Spruijt describes the measurement and analysis of the surface charge of biomolecular condensates. They use carefully constructed microchannels with electrodes to apply an electric field to a suspension of condensates. They then track their movement through microscopy and single-particle tracking. These measurements, together with the measurement of the radius of the condensate, allow the zeta potential to be measured at a single particle level. With a few exceptions, this property has previously only been measured at an ensemble level for such systems. With single particle measurements, the authors show that the assumption that the particle mobility is size-independent doesn't hold, and in fact it seems that there is a linear dependence between the size of the condensate and the electrophoretic mobility.

This is a well written paper on an important topic. I have a few minor points for the authors to consider

- The authors use condensate and coacervate as equivalent terms? Coacervates normally form from two oppositely charged polymers, while condensates can have a more diverse range of building blocks.
- Single particle mobility measurements of condensates have been performed before, e.g. in *Nano Lett.* 2022, 22, 2, 612–621. The authors may wish to connect with such work.
- The authors mention in their concluding paragraph a connection with alpha-synuclein aggregation. This is potentially interesting, but I don't get what the authors mean or what the biological implications (if any) might be.
- More generally the mechanistic implications from the authors' study are highly interesting and imply a role for molecular mobility within the condensates. Perhaps the authors could test this with systems which are able to gel.

Reviewer #2:

Remarks to the Author:

In this work, the authors address the important issue of surface charge characteristics of biomolecular condensates with improved resolution compared to previous efforts. These surface properties are important factors in determining the interactions and properties of condensates including the interfacial molecular exchange and ultimately function, however much remains to be understood about these features. Previous studies have interrogated the surface charge of condensates, primarily through laser doppler electrophoresis, which has provided limited information. Along these lines, the authors present a different approach, using microelectrophoresis and single particle tracking. The improved methodology provides new information, including that the condensates can have a size dependent zeta potential, that surface accumulation of charged species (polylysine in the example studied) then promotes interactions with other macromolecules bearing opposite charge, and that ATP can invert the surface charge. Together, these insights and the methodology will be of very good interest to the community of condensates biophysicists and biologists. However, I have some issues for the authors to address as listed below.

In their measurements of viscosity, the authors use diffusion measurements of dextran using RICS. Since the effective viscosity can vary for smaller particles (relative to an effective network mesh size, e.g., see Schuler et al. *Nature* 2023, pp876), can the authors comment/make a note about how this issue factors in for the dextran size used in this study.

Given the relatively complex nature of the studied coacervates vs simpler liquid systems such as those considered in the Ohshima et al. model used here, can the authors comment on the considerations of surface vs internal charge in the droplets and how this issue figures in the model

and analysis.

At the bottom of page 8, the authors note that "It is interesting to see how simply changing the relative concentrations of components in a coacervate can reverse their surface charge, and thereby affect their affinity for membranes and certain surface-localizing proteins." which is very nicely demonstrated by the author's detailed measurements. The authors might want to cite previous demonstrations of this charge reversal behavior (e.g., Banerjee et al. *Angew Chem.* 2017, 11354) albeit with the substantial limitations in the previous methods used as noted by the authors.

The ATP results discussed are interesting. I would suggest the authors can consider adding a couple more references to prior work on ATP (e.g., Kang et al. *Biochem Biophys Res Commun* 2018, 545; Hyman et al. *Science* 2017, 753) and condensates and potentially discussing their results in this context.

The differences between the synuclein and ATP addition to complex coacervates are interesting. For the former, addition of synuclein results in loss of mobility then no further changes with further addition. For the latter, further addition results in reversal of mobility sign. Can the authors elaborate on their interpretation.

Reviewer #3:

Remarks to the Author:

Dear Editor

The manuscript "probing the surface charge of condensates using microelectrophoresis" by van Haren et al. provides a new experimental platform to measure electrophoretic motion of microdroplets in aqueous two-phase suspensions. Their platform reveals size-dependent electrophoretic mobility for several complex coacervates and biomolecular condensates, clearly demonstrated through brightfield microscopy. This size-dependent mobility is used for precise calculation of zeta-potential, afforded by recasting previous theory used to describe electric field-induced motion of mercury droplets in KCl solution. They present zeta-potential measurements for a range of complex coacervates and model biomolecular condensates, revealing dependencies on system composition, both in terms of the ratios of polyelectrolytes and salt concentration. In particular, they show that the surface potential of K₁₀/D₁₀ coacervates is modulated by ATP, switching the sign of the zeta potential at biologically relevant mM concentrations. They also demonstrate the ability of 3 mM ATP to remove alphasynuclein adsorbed to the surface of pLys/pGlu, with an accompanying change in electrophoretic mobility, an exciting development of high interest to those studying biomolecular condensate-associated pathologies.

Their electrophoretic mobility measurements, relying particle tracking from clean microscopy images are to a high standard. The subsequent calculation of zeta potential uses particle velocities and sizes from these images, along with the Debye length, measurement of microdroplet viscosities, and quantification of the effective electric field strength. The authors clearly explain their derivation and motivation for their formula for the zeta potential, which to me seems very smart. However, I believe the authors could have been more thorough with some of the measurements used to calculate zeta potential.

In terms of novelty and impact, the platform they have presented marks a novel experimental advance in the field. This will likely be picked up by numerous groups, owing to the simplicity of their setup and clarity of its presentation. They clearly explain the motivation for this setup, and the detailed derivation of their equation for zeta potential in the supplementary information clearly highlights the merits of its' validity. It would have been nice to see a description of the relationship between zeta potential and surface charge. I would like to draw the author's attention to a study from 2022 in *ACS nanoletters* by Welsh et al. (<https://doi.org/10.1021/acs.nanolett.1c03138>). This study also performs microelectrophoresis on microdroplets of biomolecular condensates, albeit using a complex microfluidics setup, also revealing an increased surface charge inhibits droplet

fusion akin to the manuscript here. Due to the strong overlap between the study of Welsh et al. and the work presented here, I was surprised not to see the work of Welsh et al. discussed here.

Whilst the majority of the paper is well presented, and the study well thought out, I do have some reservations with the quality of data and conclusions drawn. I also feel the writing could be improved, both in terms of language and referencing. I have divided my specific comments into minor and major. Addressing my minor comments will help improve the standard of the article, aiding the reader and clearing up any possible discrepancies. My major comments are the issues which cause me to doubt the accuracy of their measurements and scientific conclusions of the work.

Minor comments

1. The authors should spend some time improving the referencing throughout. Please provide references for the appendix Equations E1 and E2. The sentence finishing on line 303 should reference DOI: 10.1126/science.abg7071. The sentence starting "compared to other studies..." on line 197 should reference other studies. The sentence finishing "...lower than previously reported" On line 201 should reference previous reports, some quoted values from previous reports would be helpful.
2. Please Explain all lines of best fit in the accompanying Figure caption.
3. Would like calculations to be more transparent, e.g. the calculation of ionic strength does not seem correct to me. It is not obvious to me how the authors reach a value of 97 mM in line 10 of the Appendix "particle electrophoresis and calculation of kR ". Why treat the polyelectrolytes as chains of monovalent ions? Is this due to the hydronium/hydroxide counterions formed upon protonation/deprotonation? Will these not be mopped up by the Tris buffer? Would be preferable for authors to appropriately reference this section of the appendix.
4. Given the phase separated nature of their two-phase suspensions, is it appropriate to consider the ionic strength as calculated assuming a homogenous solution?
5. Missing data; 4th and 5th columns in table S1 are incomplete.
6. In Figure 4c why do the authors not present zeta potentials of pLys/pGlu in the presence of alpha-synuclein and ATP? I notice that the trend for pLys/pGlu +1 μ M asyn + 3 mM ATP is distinct from all others presented in this paper – the droplets move slower with increasing radius. Why is this? I would like to see a discussion of this observation in the paper.
7. I do wonder why they use the root-mean-square displacement to calculate average velocity, is this the velocity used in calculation of zeta potentials? Shouldn't only components of velocity parallel to the applied field be used? In the video they present, droplets display Brownian motion and collision without fusion. Have they accounted for both of these factors when calculating the appropriate velocity for zeta potential analysis?
8. What is the utility of the fluorescence images in 3a and 4d?
9. How did they calculate the radial fluorescence intensities in figure 4f? What is the origin of the error bars in the plots if the data is simply a cross section? I assume the measurement is not a radial average, otherwise the data would be symmetric about the droplet centroid as presented.
10. I don't understand figure S7. Why is there no correlation in the y direction? Because the line scan is slow (10 Hz)?
11. Line 116 in main text. This is not obvious from the surrounding text. I see you explain this point in the introduction, but a statement as 'in fact, ...' suggests to me this is a new piece of information.
12. Did you obtain the same zeta potential using your microelectrophoresis setup for the charged polystyrene beads (lines 177-179 and Figure S9) as that measured using a standard zetasizer?

Major specific comments

1. There are a lot of estimated viscosity values in Table S2, with little justification. The value of 84.04 mPa s has been used for four different species, but the authors present no rationale for this value. For other values, the authors seem to have interpolated between values measured for coacervates/condensates formed in systems of similar compositions. The authors do not explain their rationale or reasoning for this. What method of interpolation was used? Why were the values not directly measured?
2. In the supplementary section "measurement of coacervate zeta potential by

microelectrophoresis" the authors report a threshold electric field strength is required for their droplets to move, in all instances. They attribute this to a weak adhesion to the passivated slide surface. They treat this as a static friction, transforming the applied field into a resultant field, which is given by the difference of the applied field and the threshold field. However, it is not unusual to have a dynamic friction of lower magnitude than the static friction (see <https://doi.org/10.1021/acs.langmuir.2c00178>). Maybe I'm missing something, but I would like to see at least a discussion of this, possibly experimental verification.

3. The conclusion that their equation properly accounts for Debye length is based on measuring the K30/D30 in the presence of an additional 300 mM NaCl (lines 158 to 165). They assume (Table S2) that this leaves the viscosity unchanged and they conclude the zeta potential is also unchanged. I am not convinced. Changing the salt concentration will alter the coacervate composition and viscosity (see <https://doi.org/10.1016/j.cis.2016.08.010>), likely also changing the surface charge.

4. For their measurement of coacervate viscosities, they use (in most cases) a fluorescently labelled polymeric dextran. Have the authors tested this molecules action as an inert probe? Previous studies have shown the viscosity of proteinaceous condensates to be modulated by the presence of polymeric clients (<https://doi.org/10.1016/j.molcel.2015.09.017>, <https://doi.org/10.1073/pnas.1504822112>). To test this, their viscosity measurements could be repeated as the amount of probe is altered in the sample. In the 'dilute' limit, the viscosity is not expected to be a function of probe concentration.

5. In lines 387 to 389 the authors describe the calibration used for determining condensate viscosity by Raster Image Correlation Spectroscopy (RICS). They calibrate their focal volume by imaging Alexa 488 in water. However, the refractive index of coacervates and condensates is not expected to be the same as water, owing to the high local concentration of proteins. See for example doi: 10.1529/biophysj.103.030072, <https://doi.org/10.1101/2020.10.25.352823> and <https://doi.org/10.1016/j.bpj.2011.03.004>. For their confocal imaging, this will introduce a spherical aberration when imaging in coacervates/condensates which is not controlled for by their calibration measurement in water. Are the authors able to address this? Is it possible to back up any of their viscosity measurements with an alternative method, by e.g. tracking the diffusive motion of colloidal particles immersed in coacervate/condensates?

► *Reviewer's comments are greyed out and italicized; authors' responses are in black font right below each group of comments, as well as highlighted in the manuscript in yellow.*

Reviewer #1 (Remarks to the Author):

The manuscript by van Haren and Spruijt describes the measurement and analysis of the surface charge of biomolecular condensates. They use carefully constructed microchannels with electrodes to apply an electric field to a suspension of condensates. They then track their movement through microscopy and single-particle tracking. These measurements, together with the measurement of the radius of the condensate, allow the zeta potential to be measured at a single particle level. With a few exceptions, this property has previously only been measured at an ensemble level for such systems. With single particle measurements, the authors show that the assumption that the particle mobility is size-independent doesn't hold, and in fact it seems that there is a linear dependence between the size of the condensate and the electrophoretic mobility.

This is a well written paper on an important topic. I have a few minor points for the authors to consider.

► We appreciate and share the reviewer's enthusiasm about this work. We made adjustments following each comment to further improve paper's clarity and presentation.

The authors use condensate and coacervate as equivalent terms? Coacervates normally form from two oppositely charged polymers, while condensates can have a more diverse range of building blocks.

► While it is true that many reported examples of coacervates are formed from two oppositely charged polymers, this is not strictly necessary. *Complex coacervates* can be made from any combination of oppositely charged molecules, and examples include peptides, oligonucleotides, small molecules such as citrate, malonic acid and NADH, multivalent ions such as ferricyanide and pyrophosphate, and large proteins. In addition, *simple coacervates*, which form from a single (peptide, polymer, nucleic acid) species, have also been reported, and their formation follows the same rules of LLPS as complex coacervates and liquid-like condensates. The term coacervate (simple and complex) originated in colloid science (Bungenberg de Jong, 1929) and was later adopted by Oparin in his theory on the origin of life, and is therefore often used in the fields of protocells, polymer chemistry and materials science.

On the other hand, condensate is a term that was (re)introduced in 2017 to describe non-membrane-bound inclusion bodies in cells and in vitro reconstituted droplet models of such cellular inclusion bodies (Banani et al, Nature Rev. Mol. Cell Biol. 2017). Cellular condensates typically contain a specific set of disordered proteins and nucleic acids as scaffolds, complemented with various other proteins, peptides and small molecules as guests or client molecules. It is usually assumed that condensates form by liquid-liquid phase separation and in that case *there is no fundamental difference between a liquid-like condensate and a coacervate*. Condensates have been shown to undergo maturation into more solid- or gel-like structures and may be more gel-like and dynamically arrested already upon formation, but also "coacervates" composed of short peptides that undergo maturation (Lipinski et al, Chem. A Eur. J. 2023;) and gel-like, arrested "coacervates" have been reported (Alshareedah et al, Nature Comm. 2021). In practice, the term condensate is more commonly used in the fields of cell and molecular biology, while coacervate is more commonly used in chemistry and colloid science, and coacervates are often composed of "simpler" building blocks with a single or limited number of interaction types between the building blocks. Condensates (especially the cellular inclusion bodies) are composed of more different components which also tend to have a more diverse sequence composition.

In our manuscript, we use the terms condensate and coacervate in these same contexts. Our finding that electrophoretic mobility will be size dependent holds for both (liquid-like) condensates and coacervates. Most experiments were performed with complex coacervates formed with oppositely charged peptides. We also investigated the size-dependent electrophoretic mobility of several condensate models, or in vitro reconstituted droplet systems made with cellular proteins and RNA (Figure 3). We introduced the terms condensate and coacervate in the abstract (lines 8-10) and introduction (lines 33-36). We now added an explanation of how we use these terms in our manuscript.

Single particle mobility measurements of condensates have been performed before, e.g. in Nano Lett. 2022, 22, 2, 612–621. The authors may wish to connect with such work.

► We thank the reviewer for suggesting the discussion of this work. We have now included this work in the introduction, and explain the differences between the system of Welsh *et al.* and our system.

The authors mention in their concluding paragraph a connection with alpha-synuclein aggregation. This is potentially interesting, but I don't get what the authors mean or what the biological implications (if any) might be.

► Alpha-synuclein aggregation has been linked to Parkinson's disease and its aggregation propensity is enhanced when it adsorbs to the coacervate interface (Lipinski *et al.*, Science Advances 2022). In figure 4d-4f we show how fluorescently labelled α -synuclein was completely displaced from the droplet interface and released into solution. By removing α -synuclein from the interface of phase separated liquid droplets composed of peptides its aggregation propensity could be reduced. If similar charge interactions govern the affinity of α -synuclein for the interface of cellular condensates (which remains to be verified in vivo), regulation of ATP levels may present a way in which this affinity and resulting localization can be modulated. In this manuscript, we only discuss findings with α -synuclein and ATP at the surface of coacervate droplets and suggest that these findings may be relevant for condensates inside cells as well, as we expect these condensates to have a nonzero surface charge that can potentially be modified by ATP. We do not intend to suggest direct biological implications of our findings, as that would require determining these properties in vivo. We have adapted the wording in line 284-285 to avoid any confusion about this.

More generally the mechanistic implications from the authors' study are highly interesting and imply a role for molecular mobility within the condensates. Perhaps the authors could test this with systems which are able to gel.

► We appreciate the reviewer's comments on molecular mobility within condensates. Although we did not include a system that forms a true (hydro)gel, we show that coacervates formed from an arginine-rich peptide and poly-U have a very low electrophoretic mobility at 0 M NaCl and a high mobility at 0.75 M NaCl (figure 3b). At 0 M NaCl, these coacervates are more gel-like, having longer (shear) relaxation timescales and higher zero-shear viscosities, while at 0.75 M NaCl these droplets behave more liquid-like at typical timescales of the motion (see Lui *et al.*, Advances in Colloid and Interface Science 2017). These results underscore the role of the molecular mobility within the condensates the reviewer is referring to. We added a statement to the discussion of the results in figure 3b to highlight this.

Reviewer #2 (Remarks to the Author):

In this work, the authors address the important issue of surface charge characteristics of biomolecular condensates with improved resolution compared to previous efforts. These surface properties are important factors in determining the interactions and properties of condensates including the interfacial molecular exchange and ultimately function, however much remains to be understood about these features. Previous studies have interrogated the surface charge of condensates, primarily through laser doppler electrophoresis, which has provided limited information. Along these lines, the authors present a different approach, using microelectrophoresis and single particle tracking. The improved methodology provides new information, including that the condensates can have a size dependent zeta potential, that surface accumulation of charged species (polylysine in the example studied) then promotes interactions with other macromolecules bearing opposite charge, and that ATP can invert the surface charge. Together, these insights and the methodology will be of very good interest to the community of condensates biophysicists and biologists. However, I have some issues for the authors to address as listed below.

► We appreciate the reviewer's remarks and thank them for the helpful questions, which we address one by one below.

In their measurements of viscosity, the authors use diffusion measurements of dextran using RICS. Since the effective viscosity can vary for smaller particles (relative to an effective network mesh size, e.g., see Schuler et al. Nature 2023, pp876), can the authors comment/make a note about how this issue factors in for the dextran size used in this study.

► We thank the reviewer for pointing out the importance of condensate mesh-size. We are indeed aware that the effective network mesh size is relevant for the viscosity of a probe experience, which is dependent on the size of the probe, as illustrated nicely by Holyst and coworkers.

Our probe (1.4 nm) is near the expected mesh size of 2-3 nm for pLys/pGlu complex coacervates (McCall et al, Biophysical Journal 2018) or 1.2 nm for gelatin coacervates (Singh et al, International journal of biological macromolecules 2007), and therefore we expect that measurements with larger probes would yield similar viscosities (i.e., the “macroscopic” viscosity).

However, to obtain high-quality RICS data required for accurate viscosity determination, a probe that partitions into the droplet phase is required. Very large probes partition less or not at all in the coacervate phase (Schuster et al., Nature Comm. 2018). In this manuscript, we made a trade-off between probe partitioning and probe size that still characterizes the network viscosity, which resulted in Dex(4.4k) as probe.

To verify the assumption that our probe is sufficiently large to probe the network-level viscosity, we have performed an additional RICS experiment on K₁₀₀/D₁₀₀ with FITC-Dex(10k) instead of TRITC-Dex(4.4k), which shows a similar observed viscosity (504 mPa s for 10k and 559 mPa s for 4.4k). These results are included in Table S1.

Given the relatively complex nature of the studied coacervates vs simpler liquid systems such as those considered in the Ohshima et al. model used here, can the authors comment on the considerations of surface vs internal charge in the droplets and how this issue figures in the model and analysis.

► One important assumption in the theoretical work by Ohshima is that “no electrolyte ions can penetrate the drop surface, which implies that no electrochemical reactions occur on the drop surface, i.e. neither neutralization of ions nor ionization of atoms of the drop takes place. This assumption is equivalent to postulating that a []drop behaves like an ideally polarizable conductor. Consequently we may assume that neither electrostatic charge, nor field nor current exists inside the drop and that the drop surface is always equipotential”. This was stated in the appendix.

Complex coacervate droplets consist of ionic species, so the assumption that no electrolyte ions can penetrate the drop surface is not valid. However, coacervates can be seen as polarizable conductors (Agrawal et al, PNAS 2022). Additionally, we can assume that the electrostatic potential inside of a coacervate droplet is very close to zero, as opposite charges and counterions are present at such high concentrations that any excess (local) charge is effectively neutralized (Majee et al, Arxiv 2023).

At the bottom of page 8, the authors note that "It is interesting to see how simply changing the relative concentrations of components in a coacervate can reverse their surface charge, and thereby affect their affinity for membranes and certain surface-localizing proteins." which is very nicely demonstrated by the author's detailed measurements. The authors might want to cite previous demonstrations of this charge reversal behavior (e.g., Banerjee et al. Angew Chem. 2017, 11354) albeit with the substantial limitations in the previous methods used as noted by the authors.

► We thank the reviewer for pointing out previous research on charge inversion. We added this reference at line 228.

The ATP results discussed are interesting. I would suggest the authors can consider adding a couple more references to prior work on ATP (e.g., Kang et al. Biochem Biophys Res Commun 2018, 545; Hyman et al. Science 2017, 753) and condensates and potentially discussing their results in this context.

► We share the reviewer’s interest for ATP. The molecule seems to have more functions than just providing energy for countless cellular processes. We added the notion of ATPs recent discovered hydrotropic behaviour at lines 288-290, including new references.

The differences between the synuclein and ATP addition to complex coacervates are interesting. For the former, addition of synuclein results in loss of mobility then no further changes with further addition. For the latter, further addition results in reversal of mobility sign. Can the authors elaborate on the their interpretation.

► ATP is a small molecule with three negatively charged phosphate groups and a heterocyclic adenine group, which gives the molecule a relatively high charge density compared to α -synuclein which contains a net negative charge of $-9e$ and weighs 14.5 kDa. Moreover, the concentration of ATP added is higher than the added α -synuclein, 3 mM and 10 μ M, respectively. We hypothesize that the higher charge density of ATP results in a reversal of the droplet charge upon addition, similar to K_{10}/D_{30} being positively charged at a 2:1 lysine/aspartic acid ratio and negatively charged at a 1:2 ratio.

Reviewer #3 (Remarks to the Author):

The manuscript “probing the surface charge of condensates using microelectrophoresis” by van Haren et al. provides a new experimental platform to measure electrophoretic motion of microdroplets in aqueous two-phase suspensions. Their platform reveals size-dependent electrophoretic mobility for several complex coacervates and biomolecular condensates, clearly demonstrated through brightfield microscopy. This size-dependent mobility is used for precise calculation of zeta-potential, afforded by recasting previous theory used to describe electric field-induced motion of mercury droplets in KCl solution. They present zeta-potential measurements for a range of complex coacervates and model biomolecular condensates, revealing dependencies on system composition, both in terms of the ratios of polyelectrolytes and salt concentration. In particular, they show that the surface potential of K10/D10 coacervates is modulated by ATP, switching the sign of the zeta potential at biologically relevant mM concentrations. They also demonstrate the ability of 3 mM ATP to remove alpha-synuclein adsorbed to the surface of pLys/pGlu, with an accompanying change in electrophoretic mobility, an exciting development of high interest to those studying biomolecular condensate-associated pathologies.

Their electrophoretic mobility measurements, relying particle tracking from clean microscopy images are to a high standard. The subsequent calculation of zeta potential uses particle velocities and sizes from these images, along with the Debye length, measurement of microdroplet viscosities, and quantification of the effective electric field strength. The authors clearly explain their derivation and motivation for their formula for the zeta potential, which to me seems very smart. However, I believe the authors could have been more thorough with some of the measurements used to calculate zeta potential.

In terms of novelty and impact, the platform they have presented marks a novel experimental advance in the field. This will likely be picked up by numerous groups, owing to the simplicity of their setup and clarity of its presentation. They clearly explain the motivation for this setup, and the detailed derivation of their equation for zeta potential in the supplementary information clearly highlights the limits of its' validity. It would have been nice to see a description of the relationship between zeta potential and surface charge. I would like to draw the author's attention to a study from 2022 in ACS nanoletters by Welsh et al. (<https://doi.org/10.1021/acs.nanolett.1c03138>). This study also performs microelectrophoresis on microdroplets of biomolecular condensates, albeit using a complex microfluidics setup, also revealing an increased surface charge inhibits droplet fusion akin to the manuscript here. Due to the strong overlap between the study of Welsh et al. and the work presented here, I was surprised not to see the work of Welsh et al. discussed here.

► We thank the reviewer for pointing out this previous platform used for the microelectrophoresis of biomolecular condensates and agree it should have been included, as also indicated by Reviewer #1. We have adjusted lines 81-85 to include this work into the introduction, mainly stating the difference between the system of Welsh et al. and our system.

Whilst the majority of the paper is well presented, and the study well thought out, I do have some reservations with the quality of data and conclusions drawn. I also feel the writing could be improved, both in terms of language and referencing. I have divided my specific comments into minor and major. Addressing my minor comments will help improve the standard of the article, aiding the reader and clearing up any possible discrepancies. My major comments are the issues which cause me to doubt the accuracy of their measurements and scientific conclusions of the work.

Minor comments

1. The authors should spend some time improving the referencing throughout. Please provide references for the appendix Equations E1 and E2. The sentence finishing on line 303 should reference DOI: [10.1126/science.abg7071](https://doi.org/10.1126/science.abg7071). The sentence starting “compared to other studies...” on line 197 should reference other studies. The sentence finishing “...lower than previously reported” On line 201 should reference previous reports, some quoted values from previous reports would be helpful.

► We agree with the reviewer that more references could be added to the sections indicated. We added references Smoluchowski M 1903 & Berg JC 2010 to equations E1 and E2 in the appendix, respectively. Folkmann A *et al.* 2021 was added at the end of line 303 (now line 313). Additional references, as also indicated by the other Reviewers, have been added throughout the paper.

2. Please Explain all lines of best fit in the accompanying Figure caption..

► The best fits have been added to the figure captions. Generally, all linear fits relations are fitted with $mx + b$, while zeta potential distributions were described using a Gaussian probability density function, accompanied with a cofactor so raw data can be presented without normalization: $(C/(\sigma \cdot (2\pi)^{0.5} \cdot \exp(-0.5((x-\mu)/\sigma)^2)))$, where C is the cofactor, σ the standard deviation and μ the mean. We have added the descriptions of these fits to the relevant figure captions and fit parameters are listed in Table S3.

3. Would like calculations to be more transparent, e.g. the calculation of ionic strength does not seem correct to me. It is not obvious to me how the authors reach a value of 97 mM in line 10 of the Appendix "particle electrophoresis and calculation of κR ". Why treat the polyelectrolytes as chains of monovalent ions? Is this due to the hydronium/hydroxide counterions formed upon protonation/deprotonation? Will these not be mopped up by the Tris buffer? Would be preferable for authors to appropriately reference this section of the appendix.

4. Given the phase separated nature of their two-phase suspensions, is it appropriate to consider the ionic strength as calculated assuming a homogenous solution?

► We thank the reviewer to bringing these points to our attention. We have extended the appendix by adding **E4**, which is the formula we use for calculating the ionic strength. 97 mM was calculated by $0.5 \cdot ((C_{\text{TrisH}^+} \cdot Z_{\text{TrisH}^+}^2) + (C_{\text{Cl}^-} \cdot Z_{\text{Cl}^-}^2) + (C_{\text{K}10} \cdot Z_{\text{K}10}^2) + (C_{\text{Br}^-} \cdot Z_{\text{Br}^-}^2) + (C_{\text{D}10} \cdot Z_{\text{D}10}^2) + (C_{\text{Na}^+} \cdot Z_{\text{Na}^+}^2))$. To adjust the pH of 50 mM Tris buffer to 7.4, 42 mM HCl was added, so C_{TrisH^+} and C_{Cl^-} were taken as 42 mM and a concentration of 5 mM $\text{K}_{10}/\text{D}_{10}$ was used in monomer unit concentrations, so $C_{\text{K}10}$ and $C_{\text{D}10}$ are 0.5 mM. From this we get $0.5 \cdot ((42 \cdot 1) + (42 \cdot 1) + (0.5 \cdot 10^2) + (5 \cdot 1) + (0.5 \cdot 10^2) + (5 \cdot 1)) = 97$ mM. As can be seen, the polyelectrolytes were not actually "divided" into monovalent ions, but their counterions (Na^+ and Br^-) were explicitly taken into account.

We assumed that the ionic strength difference between coacervate samples with different lengths of K_n/D_n was negligible, and therefore we used the same value of κ for all samples with the same composition. We agree with the reviewer that we should have experimentally validated this assumption. We therefore adjusted our method of determining the effective ionic strength. To confirm that all samples with similar compositions have the same ionic strength, we measured the current passing through different solutions of K_n/D_n coacervates, and calibrated the current strength with NaCl solutions of known ionic strength (**Figure S1**). **Figure S1a** shows how the current depends on the ionic strength of NaCl solutions, with increasing C_{ion} . In **Figure S1b** we show that all nine combinations of K_n/D_n (with $n = 10, 30$ and 100) have similar currents. From this we can conclude that only Tris buffer and counter ions released from polylysine and polyaspartate contribute significantly to the ionic strength, and that most of the K_n and D_n are complexed and do not contribute to the ionic strength of the continuous solution. For other samples (**Figure 3**), we therefore assume similarly that the ionic strength is determined by the buffer ions and counterions released upon complexation.

Concretely, this means that the buffer contributes 41 mM TrisHCl and 9 mM Tris base (according to a pKa of 8.06), and counterions contribute 5 mM Na^+ and Br^- for K_n/D_n , which gives a C_{ion} of 46 mM. For RRP/polyU coacervates, we consider 41 mM TrisHCl, 6.2 mM counterions and 750 mM NaCl, which gives a C_{ion} of 797.2 mM. For K_{72}/ATP coacervates, we consider 50 mM HEPES to contain 22.1 mM HEPES acid and 27.9 mM zwitterionic HEPES, and 8.7 mM counterions (based on 133 charged groups in K_{72} , Nakashima *et al. Nature Communications* **2021**). The zwitterionic HEPES does not contribute to ionic strength (Stellwagen *et al., Anal Biochem* **2008**, and **Figure S1b**), so a C_{ion} of 30.8 mM was found. For NPM1/polyU coacervates, we consider 2.4 mM counterions (106 charged groups NPM1) and 250 mM potassium

glutamate together with 20 mM Tris pH 7.2 gives a C_{ion} of 267.6 mM, again considering the zwitterionic form of glutamate does not contribute to ionic strength. Lastly, for the protamine/NADH coacervates we find 8.4 mM counterions (21 charged groups in protamine, Hong Y et al, *Nature Communications* **2022**) and 50 mM Tris pH 7.4, resulting in a C_{ion} of 49.6 mM.

5. *Missing data; 4th and 5th columns in table S1 are incomplete.*

► Values are now added to the corresponding columns in table S1.

6. *In Figure 4c why do the authors not present zeta potentials of pLys/pGlu in the presence of alpha-synuclein and ATP? I notice that the trend for pLys/pGlu +1 μ M α syn + 3 mM ATP is distinct from all others presented in this paper – the droplets move slower with increasing radius. Why is this? I would like to see a discussion of this observation in the paper.*

► We thank the reviewer for pointing this out. We assume that the reason for droplets to no longer follow the trend of increasing mobility with increasing size is that there are still small patches of α -synuclein present on the droplet surface. As seen in figure 4e, some (especially the larger) droplets still have a slight halo around them, indicating that there could still be some labelled protein present, possibly because it has already assembled or aggregated into oligomers that are more difficult to remove. It is possible that α -synuclein oligomers form more readily at the surface of larger coacervate droplets, because of their larger radius of curvature (higher local ion concentration in the curved diffuse double layer), their denser packing of polypeptides at the interface, or their larger contact area with the underlying solid surface. In such cases, larger droplets would be more likely to retain α -synuclein oligomers after ATP addition, and consequently, have a lower apparent surface charge. Unfortunately, we are not able to calculate the ζ -potential for these droplets because the trend observed does not allow for application of equation 2. We added a discussion of this anomaly and our hypothesis in lines 303-308

7. *I do wonder why they use the root-mean-square displacement to calculate average velocity, is this the velocity used in calculation of zeta potentials? Shouldn't only components of velocity parallel to the applied field be used? In the video they present, droplets display Brownian motion and collision without fusion. Have they accounted for both of these factors when calculating the appropriate velocity for zeta potential analysis?*

► The root-mean-square displacement was originally used to calculate the movement of the coacervate value. Later we changed this to only use the y-displacement, though this was not corrected in the methods description. We have now corrected the methods to state: 'The mean velocity of the droplets was calculated from the droplet traces by taking the difference of the y coordinate from the centroid of the droplet between the first and last recorded position and dividing it by the elapsed time' on lines 420-421.

8. *What is the utility of the fluorescence images in 3a and 4d?*

► The microscopy images for each coacervate composition have been added for the reader to see how the samples looked under the microscope. The samples in 3a and 3d contain GFP labelled K₇₂ and NADH, respectively, which are fluorescent. Since the coacervate compositions contain fluorescent molecules we showed the fluorescence images instead of the bright field.

9. *How did they calculate the radial fluorescence intensities in figure 4f? What is the origin of the error bars in the plots if the data is simply a cross section? I assume the measurement is not a radial average, otherwise the data would be symmetric about the droplet centroid as presented.*

► We agree that this was not clear from the figure description. The data is indeed not a radial average but presents the α -synuclein fluorescence intensity cross-section of five equally sized droplets ($r = 1.5 \mu\text{m}$) with the shaded error being the standard deviation in fluorescence intensity. We have adjusted the figure caption of 4f to include this.

10. I don't understand figure S7. Why is there no correlation in the y direction? Because the line scan is slow (10 Hz)?

► The reviewer is correct that we do not see a correlation in the y-direction because the line scan is slow. We do observe a substantial correlation in the x-direction, along the scanning path ξ . When we return to the $x_0, y + 1$ position, molecules have diffused, and the autocorrelation is reduced to background level.

11. Line 116 in main text. This is not obvious from the surrounding text. I see you explain this point in the introduction, but a statement as 'in fact, ...' suggests to me this is a new piece of information.

► 'In fact' has been removed to avoid confusion. Additionally, we have added a reference to (Rashidi et al, 2021) which is a review that explains the difference between electrophoresis of liquid droplets and hard colloids.

12. Did you obtain the same zeta potential using your microelectrophoresis setup for the charged polystyrene beads (lines 177-179 and Figure S9) as that measured using a standard zetasizer?

► We repeated these measurements to determine the velocity at more values of the applied electric field in order to obtain a more accurate estimate of the ζ -potential. The ζ -potential of these particles was measured as -6.7 ± 0.6 mV with a standard zetasizer ($n = 3$), while with our method we obtained -6.0 ± 1.2 mV (when 3.75 V cm^{-1} is applied), or -6.0 ± 0.1 mV when we take the mean of the measured ζ -potentials at different electric field strengths. This value is slightly lower than the one measured on a zetasizer, but within experimental uncertainty. This was added to lines 185-189.

Major specific comments

1. There are a lot of estimated viscosity values in Table S2, with little justification. The value of 84.04 mPa s has been used for four different species, but the authors present no rationale for this value. For other values, the authors seem to have interpolated between values measured for coacervates/condensates formed in systems of similar compositions. The authors do not explain their rationale or reasoning for this. What method of interpolation was used? Why were the values not directly measured?

► We agree that reporting directly measured values would have been more accurate. We have now measured viscosities of all systems for which estimates were previously used: K_{10}/D_{30} , K_{10}/D_{100} , K_{30}/D_{10} , K_{30}/D_{100} , K_{100}/D_{10} , K_{100}/D_{30} , and K_{30}/D_{30} with 0.30 M NaCl. These new values have been reported in Table S1 and S2. All ζ -potentials have been recalculated using the new values.

We still use interpolation to estimate the viscosity of K_{10}/D_{10} coacervates with added ATP for the $[\text{ATP}] = 1 \text{ mM}$ and 2 mM by assuming a linear relation between the viscosity of K_{10}/D_{10} coacervates without ATP (37 mPa s) and K_{10}/D_{10} coacervates containing 3 mM ATP (75 mPa s). We have added an explanation of the interpolation method to the relevant captions. In these cases, the precise value of the viscosity will not alter the main finding of the ATP experiment, that ATP can reverse the surface charge of coacervates.

2. In the supplementary section "measurement of coacervate zeta potential by microelectrophoresis" the authors report a threshold electric field strength is required for their droplets to move, in all instances. They attribute this to a weak adhesion to the passivated slide surface. They treat this as a static friction, transforming the applied field into a resultant field, which is given by the difference of the applied field and the threshold field. However, it is not unusual to have a dynamic friction of lower magnitude than the static friction (see <https://doi.org/10.1021/acs.langmuir.2c00178>). Maybe I'm missing something, but I would like to see at least a discussion of this, possibly experimental verification.

► We have measured the force required moving droplets of three sizes (3, 6 and 11 μm) over the pLL-g-PEG passivated glass surface using optical tweezers. We can see that there is a static friction (threshold force required) for droplets to move, also known as the pinning force, which depends by approximation linearly on the droplet size. The drag force increases when dragging the droplets faster, which is caused by the dynamic friction. We account for the static friction by transforming the applied field to a resultant

field using a threshold electric field. The dynamic friction is taken into account in the derivation of Eq.2. When droplets move during electrophoresis and the voltage is lowered back to E_0 , droplets abruptly stop moving, indicating that the static friction is continuously present.

3. The conclusion that their equation properly accounts for Debye length is based on measuring the K_{30}/D_{30} in the presence of an additional 300 mM NaCl (lines 158 to 165). They assume (Table S2) that this leaves the viscosity unchanged and they conclude the zeta potential is also unchanged. I am not convinced. Changing the salt concentration will alter the coacervate composition and viscosity (see <https://doi.org/10.1016/j.cis.2016.08.010>), likely also changing the surface charge.

► We agree with the reviewer that increasing the salt concentration will lower the viscosity of the coacervates. As discussed above in our response to major specific comment 1, we now measured the viscosity of K_{30}/D_{30} coacervates in the presence of 300 mM by RICS and recalculated the ζ -potential (see Table S1). The viscosity was indeed lower (110 mPa s at 0.3 M NaCl compared to 130 mPa s at 0 M NaCl).

We added lines 169-171 to emphasize that the lowered viscosity also has an effect on the increased electrophoretic mobility, which results in a higher ζ -potential when calculated by the Smoluchowski equation. With the newly calculated ionic strength and acquired droplet viscosity, the ζ -potential of K_{30}/D_{30} droplets at 0.30 M salt is now lower than K_{30}/D_{30} coacervates without salt, although the droplet velocities are higher.

4. For their measurement of coacervate viscosities, they use (in most cases) a fluorescently labelled polymeric dextran. Have the authors tested this molecules action as an inert probe? Previous studies have shown the viscosity of proteinaceous condensates to be modulated by the presence of polymeric clients (<https://doi.org/10.1016/j.molcel.2015.09.017>, <https://doi.org/10.1073/pnas.1504822112>). To test this, their viscosity measurements could be repeated as the amount of probe is altered in the sample. In the 'dilute' limit, the viscosity is not expected to be a function of probe concentration.

► Currently, we use a total probe concentration of approximately 0.1 nM in the overall solution. With partition coefficients of circa 2 – 10, the internal probe concentration is less than 1 nM. We assume that such a low concentration of the probe inside the coacervate does not affect droplet viscosity. In the studies mentioned by the reviewer, the concentration of polymeric clients that were added were significantly higher.

To test this, we measured the diffusion coefficient of our 4.4k TRITC-dextran inside K_{100}/D_{100} droplets with a two times higher probe concentration. We found a viscosity of 590 mPa s while the viscosity of K_{100}/D_{100} coacervates with the original probe concentration was 560 mPa s. This is within experimental uncertainty.

5. In lines 387 to 389 the authors describe the calibration used for determining condensate viscosity by Raster Image Correlation Spectroscopy (RICS). They calibrate their focal volume by imaging Alexa 488 in water. However, the refractive index of coacervates and condensates is not expected to be the same as water, owing to the high local concentration of proteins. See for example doi: 10.1529/biophysj.103.030072, <https://doi.org/10.1101/2020.10.25.352823> and <https://doi.org/10.1016/j.bpj.2011.03.004>. For their confocal imaging, this will introduce a spherical aberration when imaging in coacervates/condensates which is not controlled for by their calibration measurement in water. Are the authors able to address this? Is it possible to back up any of their viscosity measurements with an alternative method, by e.g. tracking the diffusive motion of colloidal particles immersed in coacervate/condensates?

► We have not corrected for the difference in refractive index between the coacervates and water. This is common in FCS and RICS measurements within coacervates and other media with different refractive indices, although it could lead to slightly different results. However, we have reason to believe that the difference in refractive indices should not lead to substantial differences in viscosities. First, when measuring, we see no differences in diffusion coefficients determined in droplets with different sizes (between 10 to 30 micron). Secondly, since the droplet is an order of magnitude larger than the measured region within the droplet, the effect of the radius of curvature is relatively small. Lastly, since the refractive index of coacervates is expected to be relatively close to that of the supernatant with buffer and ions (e.g., a refractive index of 1.364 has been reported for FUS condensates, *Hong Y et al*, *Advanced Optical Materials* **2021**), we do not expect this to give rise to large spherical aberrations.

Reviewers' Comments:

Reviewer #1:

Remarks to the Author:

The authors have addressed my questions.

Reviewer #2:

Remarks to the Author:

The authors have done a very good job of addressing my previous comments. The insights and the methodology presented in this work will be of substantial interest and utility to the community of condensate biophysicists and biologists.

Reviewer #3:

Remarks to the Author:

I would like to thank the reviewers for consideration of my recommendations, and am entirely satisfied with the revised article, other than a few exceptions which I list below.

1. Clear description of surface charge and zeta potential

Surface charge is highlighted as the parameter of interest, and in the title, yet all experiments in the paper are geared towards measuring the zeta-potential. The paper would benefit from a sentence or two in the introduction emphasising the relationship between these two distinct concepts.

2. Referencing in introduction

I would like to see a few references added at the end of the sentence finishing on line 51. The authors are talking about ELS being usually converted into zeta potential, using the theory of Smoluchowski. There aren't references to Smoluchowski's work, nor any studies alluded to which apply the theory to ELS.

3. Dynamic friction

I believe the authors misinterpreted my second major specific comment referring to possible differences between dynamic and static friction. I do not think the dynamic friction I am referring to is accounted for in Equation 2. The friction in equation 2 is due to viscous drag, I am referring to the interaction of the droplets with the bottom surface of the sample. For droplets, which in motion have a non-uniform contact angle, the static friction is not necessarily equal to the friction experienced when in motion. I would be persuaded that this issue is minimized by having a contact angle close to 180°. In lines 104 and 105 this is alluded to, referring to Figure 1a which is a diagram. If possible, a side-on image showing the droplets wettability would be very helpful in clearing this up. Alternatively, do you calculate the same zeta potential for your coacervates presented in Figure S9, using your correction factor for the coacervates resting on the substrate? The coacervates at $z=100\ \mu\text{m}$ will not experience any friction (dynamic or static) with the substrate at the bottom of the sample cell.

► *Reviewer's comments are greyed out and italicized; authors' responses are in black font right below each group of comments, as well as highlighted in the manuscript in yellow.*

Reviewer #1 (Remarks to the Author):

The authors have addressed my questions.

Reviewer #2 (Remarks to the Author):

The authors have done a very good job of addressing my previous comments. The insights and the methodology presented in this work will be of substantial interest and utility to the community of condensate biophysicists and biologists.

Reviewer #3 (Remarks to the Author):

I would like to thank the reviewers for consideration of my recommendations, and am entirely satisfied with the revised article, other than a few exceptions which I list below.

1. *Clear description of surface charge and zeta potential.*

Surface charge is highlighted as the parameter of interest, and in the title, yet all experiments in the paper are geared towards measuring the zeta-potential. The paper would benefit from a sentence or two in the introduction emphasising the relationship between these two distinct concepts.

► We agree with the reviewer that emphasizing the difference between surface charge and ζ -potential and explaining how our results can still give insight into condensate surface charge benefits the manuscript. Therefore, we have added a short section to the introduction in which we define both parameters and explain their dependence.

2. *Referencing in introduction.*

I would like to see a few references added at the end of the sentence finishing on line 51. The authors are talking about ELS being usually converted into zeta potential, using the theory of Smoluchowski. There aren't references to Smoluchowski's work, nor any studies alluded to which apply the theory to ELS.

► We have added two references about Smoluchowski's work after line 51 (now 58), including the paper to Smoluchowski's original work (that was already included in the supplementary information).

3. *Dynamic friction*

I believe the authors misinterpreted my second major specific comment referring to possible differences between dynamic and static friction. I do not think the dynamic friction I am referring to is accounted for in Equation 2. The friction in equation 2 is due to viscous drag, I am referring to the interaction of the

droplets with the bottom surface of the sample. For droplets, which in motion have a non-uniform contact angle, the static friction is not necessarily equal to the friction experienced when in motion. I would be persuaded that this issue is minimized by having a contact angle close to 180° . In lines 104 and 105 this is alluded to, referring to Figure 1a which is a diagram. If possible, a side-on image showing the droplets wettability would be very helpful in clearing this up. Alternatively, do you calculate the same zeta potential for your coacervates presented in Figure S9, using your correction factor for the coacervates resting on the substrate? The coacervates at $z=100\ \mu\text{m}$ will not experience any friction (dynamic or static) with the substrate at the bottom of the sample cell.

► We apologize for the misunderstanding of the reviewer's previous comment. To address this point, we followed both suggestions.

When calculating the ζ -potential for K_{10}/D_{10} coacervates we find a ζ -potential of $+2.5 \pm 0.1\ \text{mV}$. For this we used the apparent E-field of $3.77\ \text{V cm}^{-1}$ ($6.17 - E_0$, where E_0 is $2.4\ \text{V}$). When we use the fitted equation to calculate the ζ -potential for the K_{10}/D_{10} droplets at $z = 100\ \mu\text{m}$ without the correction for E_0 , we find a ζ -potential of $2.8 \pm 0.97\ \text{mV}$. This result is calculated from five droplets, and therefore we can only say that the value is indeed in the same range.

Additionally, we have constructed a side-on image from K_{10}/D_{10} coacervates containing $1\ \mu\text{M}$ Cy5 where the contact angle is visible. In this image, the slide surface present at the $0,0,0$ coordinate and goes along the x and y coordinate. As can be seen in the z-direction, droplet contact angle is close to 180° . Therefore, we conclude that the dynamic interaction of the droplets with the bottom surface of the sample is minimal.

3D projection